# Variation in the ACE2 receptor has limited utility for SARS-CoV-2 host prediction

**Nardus Mollentze[1,2]\*, Deborah Keen[1], Uuriintuya Munkhbayar[1], Roman Biek[1], Daniel G Streicker[1,2]**

[1]School of Biodiversity, One Health & Veterinary Medicine, College of Medical, Veterinary, and Life Sciences, University of Glasgow, Glasgow, United Kingdom; [2]Medical Research Council – University of Glasgow Centre for Virus Research, Glasgow, United Kingdom

**Abstract** Transmission of SARS-CoV-2 from humans to other species threatens wildlife conservation and may create novel sources of viral diversity for future zoonotic transmission. A variety of computational heuristics have been developed to pre-emptively identify susceptible host species based on variation in the angiotensin-converting enzyme 2 (ACE2) receptor used for viral entry. However, the predictive performance of these heuristics remains unknown. Using a newly compiled database of 96 species, we show that, while variation in ACE2 can be used by machine learning models to accurately predict animal susceptibility to sarbecoviruses (accuracy = 80.2%, binomial confidence interval [CI]: 70.8–87.6%), the sites informing predictions have no known involvement in virus binding and instead recapitulate host phylogeny. Models trained on host phylogeny alone performed equally well (accuracy = 84.4%, CI: 75.5–91.0%) and at a level equivalent to retrospective assessments of accuracy for previously published models. These results suggest that the predictive power of ACE2-based models derives from strong correlations with host phylogeny rather than processes which can be mechanistically linked to infection biology. Further, biased availability of ACE2 sequences misleads projections of the number and geographic distribution of at-risk species. Models based on host phylogeny reduce this bias, but identify a very large number of susceptible species, implying that model predictions must be combined with local knowledge of exposure risk to practically guide surveillance. Identifying barriers to viral infection or onward transmission beyond receptor binding and incorporating data which are independent of host phylogeny will be necessary to manage the ongoing risk of establishment of novel animal reservoirs of SARS-CoV-2.

\*For correspondence:
nardus.mollentze@glasgow.ac.uk

**Competing interest:** The authors declare that no competing interests exist.

## Editor's evaluation

Many approaches to predict which animals species might be at risk of infection by SARS-CoV-2 focus on features of the ACE2 host cell receptor to which the virus binds. This important study shows that such methods are not uncovering a true biological signal. Instead, Mollentze and colleagues show that ACE2 sequences are effectively only a proxy for generic species relationships, and species phylogeny alone can provide equivalent predictive power.

## Introduction

The global spread and high incidence of severe acute respiratory syndrome-associated virus 2 (SARS-CoV-2) has enabled human-to-animal transmission (hereafter, 'reverse zoonotic' transmission) in a variety of taxa and environmental contexts. For example, reports have described household

**eLife digest** The COVID-19 pandemic affects humans, but also many of the animals we interact with. So far, humans have transmitted the SARS-CoV-2 virus to pet dogs and cats, a wide range of zoo animals, and even wildlife. Transmission of SARS-CoV-2 from humans to animals can lead to outbreaks amongst certain species, which can endanger animal populations and create new sources of human infections. Thus, careful monitoring of animal infections may help protect both animals and humans.

Identifying which animals are susceptible to SARS-CoV-2 would help scientists monitor these species for outbreaks and viral circulation. Unfortunately, testing whether SARS-CoV-2 can infect different species in the laboratory is both time-consuming and expensive. To overcome this obstacle, researchers have used computational methods and existing data about the structure and genetic sequences of ACE2 receptors – the proteins on the cell surface that SARS-CoV-2 uses to enter the cell – to predict SARS-COV-2 susceptibility in different species. However, it remained unclear how accurate this approach was at predicting susceptibility in different animals, or whether their correct predictions indicated causal links between ACE2 variability and SARS-CoV-2 susceptibility.

To assess the usefulness of this approach, Mollentze et al. started by using data on the ACE2 receptors from 96 different species and building a machine learning model to predict how susceptible those species might be to SARS-CoV-2. The susceptibility of these species had either been observed in natural infections – in zoos, for example – or had been assessed in the laboratory, so Mollentze et al. were able to use this information to determine how good both their model and previous approaches based on the sequence of ACE2 receptors were.

The results showed that while the model was quite accurate (it correctly predicted susceptibility to SARS-CoV-2 about 80% of the time), its predictions were based on regions of the ACE2 receptors that were not known to interact with the virus. Instead, the regions that the machine learning model relied on were ones that tend to vary more the more distantly related two species are. This indicates that existing computational approaches are likely not relying on information about how ACE2 receptors interact with SARS-CoV-2 to predict susceptibility. Instead, they are simply using information on how closely related the different animal species are, which is much easier to source than data about ACE2 receptors.

Indeed, the sequences of the ACE2 receptors in many species are unknown and the species for which this information is available come only from a few geographic areas. Mollentze et al. also showed that limiting the predictions about susceptibility to these species could mislead scientists when deciding which species and geographic areas to surveil for possible viral circulation.

Instead, it may be more effective and cost-efficient to use animal relatedness to predict susceptibility to SARS-CoV-2. This makes it possible to make predictions for nearly all mammals, while being just as accurate as models based on ACE2 receptor data. However, Mollentze et al. point out that this approach would still fail to narrow down the number of animals that need to be monitored enough for it to be practical. Considering additional factors like how often the animals interact with humans or how prone they are to transmit the virus among themselves may help narrow it down more. Further research is therefore needed to identify the best multifactor approaches to identifying which animal populations should be monitored.

transmission to domestic dogs and cats, transmission to lions and tigers in zoos, transmission to farmed mink, and transmission to free-living white-tailed deer (*Barrs et al., 2020*; *Kuchipudi et al., 2022*; *McAloose et al., 2020*; *Sit et al., 2020*). While the majority of such spillovers are likely to be dead-ends for the virus, repeated introductions followed by onward transmission have been reported in both European mink and North American deer (*Kuchipudi et al., 2022*; *Oreshkova et al., 2020*; *Oude Munnink et al., 2021*). The potential for reverse zoonotic transmission to both threaten wild and domestic animal health and to foster viral evolutionary diversification that could ultimately be re-introduced into humans (*Hammer et al., 2021*; *Oude Munnink et al., 2021*) has led to a surge of interest in pro-actively identifying potentially susceptible species in which surveillance could more efficiently detect reverse zoonotic spillover or sustained transmission of SARS-CoV-2.

Evidence-based prioritisation of SARS-CoV-2 surveillance in animals has proven intractable through traditional approaches. Animal infection experiments have demonstrated that a broad range of

species spanning multiple mammalian orders are susceptible (*Freuling et al., 2020*; *Munster et al., 2020*; *Mykytyn et al., 2021*; *Schlottau et al., 2020*; *Shi et al., 2020*; *Zhao et al., 2020*). Yet, infection experiments also suggest fine-scaled variation in susceptibility within mammalian orders. For example, Egyptian fruit bats (*Rousettus aegyptiacus*) can be infected experimentally with SARS-CoV-2, while attempts to establish infection in big brown bats (*Eptesicus fuscus*) have thus far failed (*Hall et al., 2020*; *Schlottau et al., 2020*). This combination of wide host range with variable susceptibility within groups creates logistical and financial challenges to establishing susceptibility using experimental infections, since large numbers of taxa would need to be investigated. Cell culture-based compatibility assays, an alternative to in vivo infections for evaluating potential host range, are more scalable, but how well they map to susceptibility remains unclear.

A large number of computational heuristics have been developed to address the challenge of predicting the host range of SARS-CoV-2 in the absence of comprehensive experimental evidence. Most utilise the amino acid sequence of angiotensin-converting enzyme 2 (ACE2, the host receptor involved in entry of SARS-CoV-2) and base predictions on measures of similarity to human ACE2, reasoning that changes relative to this sequence (particularly at amino acid positions known to interact with the SARS-CoV-2 spike protein) decrease the probability of successful infection (*Ahmed et al., 2021*; *Alexander et al., 2020*; *Damas et al., 2020*; *Frank et al., 2020*; *Kumar et al., 2021*; *Luan et al., 2020*; *Melin et al., 2020*; *Qiu et al., 2020*). A related structural approach has compared computational estimates of binding strength (here grouped together as 'binding affinity') between the modelled ACE2 protein structures of different species and the SARS-CoV-2 spike protein, reasoning that stronger binding supports susceptibility (*Huang et al., 2020*; *Lam et al., 2020*; *Rodrigues et al., 2020*). A final approach predicted whether binding strength was above a fixed threshold which was assumed to represent susceptibility and capacity for onward transmission, using ecological and physiological traits of host species (*Fischhoff et al., 2021*). A limitation common to all efforts to date is that they have focused on untested surrogates of susceptibility, namely ACE2 similarity or binding affinity. This has two immediate drawbacks. First, how these surrogates map to susceptibility is unknown, leading to the development of largely arbitrary thresholds for deciding risk of infection. Second, since no model has been trained or validated using observed data on susceptibility (the outcome of interest), standard metrics of predictive value (e.g., sensitivity, specificity, accuracy) have not been quantified.

The recent accumulation of reports of SARS-CoV-2 spillover to various species, alongside ongoing efforts to characterise the host range of sarbecoviruses experimentally and via virus discovery, provides an exciting opportunity to develop verifiable models of sarbecovirus host range. Here, we assembled a database of compatibility across 96 species and trained sets of machine learning models to distinguish susceptible from non-susceptible species. We sought to understand the relative predictive power of commonly employed sequence and structural binding-based representations of ACE2 and to retrospectively assess the accuracy of previously published prediction methods against our compatibility database. By incorporating new metrics into model training, we also tested whether the exclusive focus on ACE2 is warranted, and whether predictive performance could be improved by considering evolutionary relationships among hosts. Finally, we used the best-performing models to assess geographic and taxonomic biases in the distribution of predicted sarbecovirus hosts that may arise depending on input data available and explore whether sarbecovirus host range prediction can be used to generate actionable insights for ongoing SARS-CoV-2 surveillance efforts in non-human animals.

## Results

Our final dataset contained records of interactions between sarbecoviruses and 96 mammalian and avian host species. Reports included natural infections (N=25) and the outcomes of experimental inoculations of captive animals (N=28), where in both cases subsequent detection of viral RNA in, or virus isolation from, nasal or rectal swabs or faeces was considered evidence of susceptibility. We also separately recorded whether species were shown to shed infectious virus (either through virus isolation or infection of other animals), but available data were limited to 23 species, of which only 17 shed infectious virus (*Figure 1*). To investigate the utility of in vitro data in the absence of natural infections or in vivo data, we also collected evidence of compatibility in cell culture experiments which involved either direct inoculation of cell cultures from each species (N=4) or inoculation of

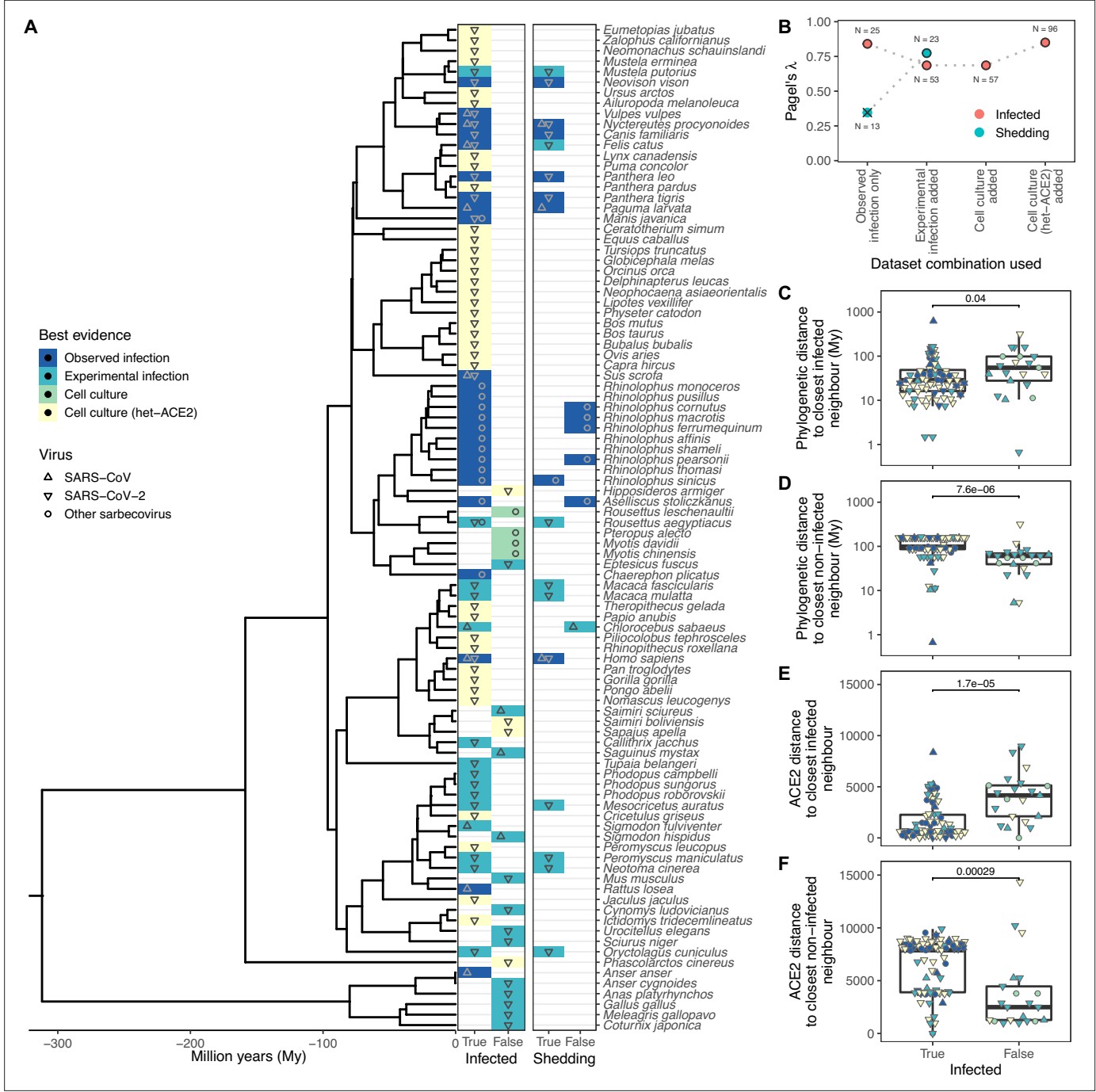

**Figure 1.** Phylogenetic clustering of sarbecovirus host susceptibility and shedding data. (**A**) Species for which susceptibility to infection and shedding of infectious virus have been assessed. Colours indicate the best available evidence, while symbols show the viruses involved. Blank rows in the shedding panel indicate missing data. A composite phylogeny derived from TimeTree indicates evolutionary relationships and median estimated divergence times. (**B**) Measurements of phylogenetic clustering when considering increasingly relaxed evidence quality thresholds, based on the phylogeny in (**A**). Outlined circles show likelihood ratio p-values ≤0.003, indicating greater clustering of infection records than expected by chance, while the cross indicates a p-value of 0.186. (C–D) Pairwise cophenetic distances between each host and its closest infected (**C**) or non-infected (**D**) neighbour. (E–F) Total Grantham distance between the angiotensin-converting enzyme 2 (ACE2) sequence of each host and its closest infected (**E**) or non-infected (**F**) neighbour. p-Values from a Wilcoxon rank sum test are indicated in (C–F), and overlapping values from different hosts are jittered horizontally. Note that while both p-values and boxplots are based on a single value for each host, overlapping symbols at the same position are used to indicate different viruses (e.g., a species known to be susceptible to both SARS-CoV [▲] and SARS-CoV-2 [▼] would be indicated using a star formed of overlapping triangles).

*Figure 1 continued on next page*

*Figure 1 continued*

The online version of this article includes the following figure supplement(s) for figure 1:

**Figure supplement 1.** Congruence between a phylogeny reconstructed from ACE2 amino acid sequences and a consensus time-scaled phylogeny for mammals and birds obtained from the TimeTree database.

cells expressing heterologous ACE2 (N=39). We treated these different sources of information hierarchically, considering the best available evidence for compatibility or incompatibility in each host species (natural infection > experimental infection > cell culture > heterologous ACE2 cell culture experiments). Importantly, we included records from any strain of *Severe Acute Respiratory Syndrome Coronavirus*, a species which includes SARS-CoV, SARS-CoV-2, and several related strains. Therefore, a key assumption of our work is that susceptibility to different sarbecoviruses overlaps sufficiently to be modelled jointly. In support of this assumption, most susceptible species in our data (88%) have been linked to at least one sarbecovirus known to use the ACE2 receptor, and among the 10 included host species with records of natural SARS-CoV-2 infection, many were also reported to be susceptible to other sarbecoviruses (N=6), primarily SARS-CoV (N=5, *Supplementary file 1*).

To explore the host range of sarbecoviruses and how susceptibility varies within animal groups, we first quantified how susceptible and non-susceptible species were distributed across both a general time-scaled phylogeny based on the accepted patterns of evolutionary divergence among hosts (derived from TimeTree; *Kumar et al., 2017*) and a maximum likelihood phylogeny constructed from ACE2 orthologs (using the NCBI reference sequence for each species in most cases, see Methods). Sarbecovirus susceptibility was highly conserved within both host phylogenies (Pagel's $\lambda$ >0.686, likelihood ratio test p-values ≤0.003; *Figure 1A–B*). Clustering was particularly strong among non-susceptible species, which tended to be closer to other non-susceptible species than known susceptible host species, both in terms of divergence dates and ACE2 amino acid distances (*Figure 1C–F*). This remained true when focusing on mammals only (Wilcoxon rank sum test p-value <0.001), indicating that results were not driven by the large evolutionary divergence between birds (mostly non-susceptible) and mammals, which comprised the majority of susceptible records. Further, a phylogeny based on all available ACE2 ortholog sequences closely matched the topology and branch lengths of the time-scaled phylogeny (Baker's γ correlation coefficient = 0.934, permutation test p-value <0.001; Spearman's $\rho$ correlation of pairwise distances = 0.870, *Figure 1—figure supplement 1*). Together, these results show that variation in ACE2 largely mirrors host evolutionary relationships, but it is still possible that current ACE2-based heuristics for predicting susceptibility to SARS-CoV-2 leverage areas of phylogenetic incongruence.

We therefore trained a series of machine learning models using metrics which captured the breadth of currently used representations of the information embedded in ACE2 (sequence dissimilarity and binding affinity with SARS-CoV-2 spike). We also tested whether representing information about individual ACE2 amino acid positions could outperform these commonly used representations. Individual positions were represented by either the observed amino acids ('AA categorical'), the physicochemical properties of observed amino acids ('AA properties'), or as a distance to the most common amino acid observed among susceptible species at the same position ('AA consensus distance'). In terms of sensitivity, no ACE2 representation performed markedly better than a null model which randomly assigned species as 'susceptible' or 'non-susceptible' in proportion to the frequency of susceptible species in the training data (*Figure 2*). In contrast, specificity varied more dramatically among representations and all outperformed the null model, indicating that ACE2 can rule out non-susceptible species. However, since non-susceptible species were rare in our training data (N=21), only two models (using either the AA consensus distance or AA properties representations) clearly outperformed the null in terms of overall accuracy. The best-performing model (AA consensus distance) correctly predicted the susceptibility or non-susceptibility of 77 of 96 species (80.2%, 95% binomial confidence interval [CI]: 70.8–87.6%). Among other models, we found that encoding ACE2 amino acid distances relative to the closest susceptible species ('distance to susceptible') outperformed the widely used metric of distance to human ACE2 ('distance to humans') in terms of specificity while maintaining high sensitivity. The AA categorical model performed considerably worse than the null across all performance metrics (*Figure 2*). Combining all representations of ACE2 had little effect on performance (74.0% accurate, CI: 64.0–82.4%), suggesting that divergence at key ACE2 amino acids from the susceptible consensus (and most other possible representations of ACE2 sequences) and

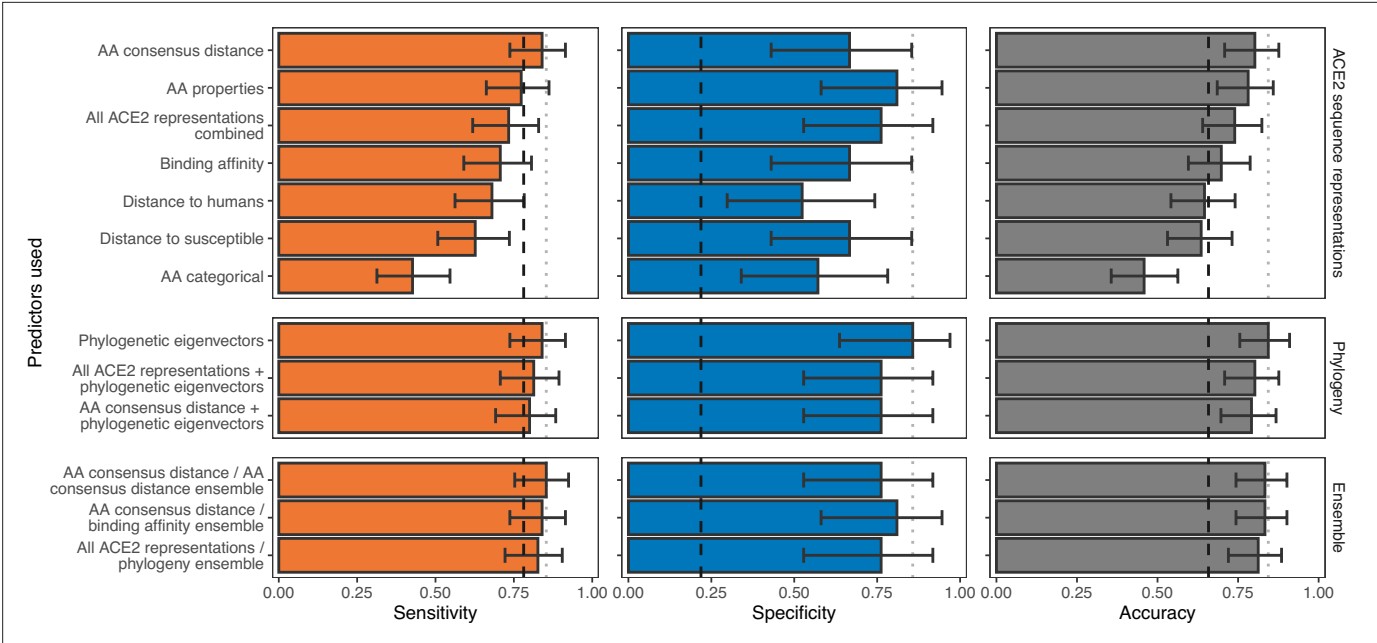

**Figure 2.** Ability of models trained on different representations of either ACE2 sequences or a time-scaled amniote phylogeny to predict host susceptibility to sarbecovirus infection. Bars represent proportions from leave-one-out cross-validation, with error bars indicating 95% binomial confidence intervals. Dashed vertical lines indicate the performance expected from a null model which randomly assigns susceptibility in proportion to its frequency in the training data (78.12% of available hosts are considered susceptible, N=96). Dotted vertical lines highlight performance of the best model in each panel.

The online version of this article includes the following figure supplement(s) for figure 2:

**Figure supplement 1.** Ability of models trained on different representations of either ACE2 sequences or a time-scaled amniote phylogeny to predict shedding of infectious virus after sarbecovirus infection.

**Figure supplement 2.** Influence of different sources of sarbecovirus susceptibility data on prediction accuracy.

**Figure supplement 3.** Performance of a model trained with all ACE2 representations on hosts linked to sarbecoviruses not known to use ACE2.

**Figure supplement 4.** Performance of a model trained with phylogenetic eigenvectors on hosts linked to sarbecoviruses not known to use ACE2.

structural models of binding affinity supply equivalent information. However, mis-classified species differed between individual models, meaning performance was improved by averaging predictions from any two ACE2-based models (ensemble models, *Figure 2*). This included an ensemble averaging predictions from two iterations of a model trained on the same ACE2 representation, suggesting that these minor improvements in performance stem purely from random variation between models (likely driven by the large numbers of features available for selection in any one representation) and not from unique information in different sequence representations (AA consensus distance/AA consensus distance ensemble, *Figure 2*).

High congruence between the overall host phylogeny and the phylogeny of ACE2 (*Figure 1*, *Figure 1—figure supplement 1*) raised the possibility that successful prediction of sarbecovirus hosts could arise through the correlation of variation in ACE2 with other evolutionarily conserved features of host biology which define susceptibility, rather than reflecting the disproportionate mechanistic importance of ACE2. Exploiting this correlation would be practically desirable since it would produce models which could be applied to all species, rather than only those with publicly available ACE2 sequences. We therefore tested models of sarbecovirus susceptibility which used phylogenetic eigenvectors to represent the time-scaled phylogeny for all amniotes (the taxonomic clade including both mammals and birds). The phylogeny-only model performed equivalently to the best ACE2-only models, correctly predicting susceptibility labels of 81 species (84.4%, CI: 75.5–91.0%) with a nearly identical sensitivity alongside improved specificity ('phylogenetic eigenvectors', *Figure 2*). Combining phylogeny with either the best-performing ACE2 representation or all ACE2 representations performed similarly to phylogeny alone (*Figure 2*), suggesting that ACE2 sequences and measures of binding affinity supply information that is redundant with host divergence patterns. Similar performance was also observed

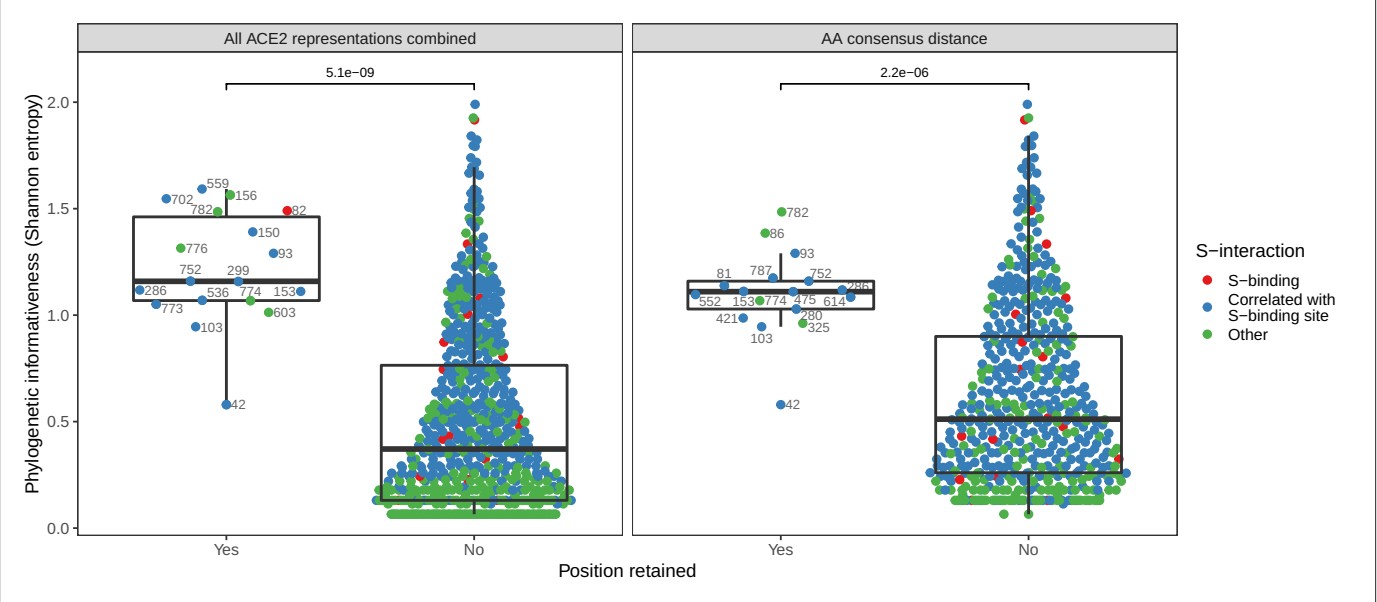

**Figure 3.** Phylogenetic informativeness of all ACE2 amino acid positions available for selection in the model. Positions are stratified by whether or not they form part of any features retained by the combined ACE2-based model (i.e., the model trained with access to all ACE2 representations, left panel) or the best-performing ACE2-based model, which lacked binding affinity information (right panel). Colours indicate whether the site in question is known to interact with the SARS-CoV-2 spike protein (based on annotations in the human reference sequence, accession number NP_001358344.1) or belongs to a cluster of correlated features which also contains at least one such spike-interacting site. Clusters were obtained using affinity propagation clustering of pairwise Spearman correlations between sites. p-Values obtained through a Wilcoxon rank sum test. Sequence positions indicated next to selected points refer to locations in the human ACE2 reference sequence. Overlapping points are jittered horizontally.

The online version of this article includes the following figure supplement(s) for figure 3:

**Figure supplement 1.** Features retained and used by the combined ACE2-based model for predicting susceptibility to sarbecovirus infection.

**Figure supplement 2.** Model performance when training the best-performing non-ensemble ACE2-only model ('AA consensus distance') with access to either all sites (as in *Figure 2*) or with representations of known SARS-CoV-2 spike-binding sites only.

**Figure supplement 3.** Model performance after training with and without data from rhinolophid bats.

when averaging numeric predictions from individual ACE2-based and phylogeny-only models ('All ACE2 representations/phylogeny ensemble', *Figure 2*; 81.2% accurate, CI: 72.0–88.5%).

Deeper analysis of the features used by our ACE2-only models to classify host susceptibility supported the hypothesis that much of their accuracy derives from correlation with host phylogeny. The model trained with all ACE2-representations (but not phylogeny) retained 25 features, of which just two were clearly linked to virus interaction (both representations of binding affinity, *Figure 3— figure supplement 1*). The remaining 23 features described information about individual ACE2 positions, primarily in the form of amino acid properties (hydrophobicity, polarity, and Van der Waals volume). Several of these features represented the same positions (e.g., hydrophobicity and Van der Waals volume of position 752), meaning the model used just 18 positions in the ACE2 alignment (*Figure 3—figure supplement 1*). Notably, only one of these positions is known to interact with the SARS-CoV-2 spike protein (position 82), but included positions did contain more phylogenetic information than average (as measured by Shannon entropy, *Figure 3*; Wilcoxon rank sum test p-value <0.001), suggesting they largely reconstructed host phylogeny. Similar results were observed in the best-performing ACE2-only model (the individual site-based 'AA consensus distance' model), which retained no known spike-binding sites (*Figure 3*). This implies that the inclusion of binding affinity measurements cannot explain the 'All ACE2 representation' model's selection of phylogenetically informative sites over known spike-binding sites. It is possible that the model-selected sites were randomly selected proxies for the same information in correlated spike-binding sites. However, it is unclear why the same sites would have been repeatedly selected from among the large number of correlated alternatives (*Figure 3*), and why they would be consistently preferred over spike-interacting

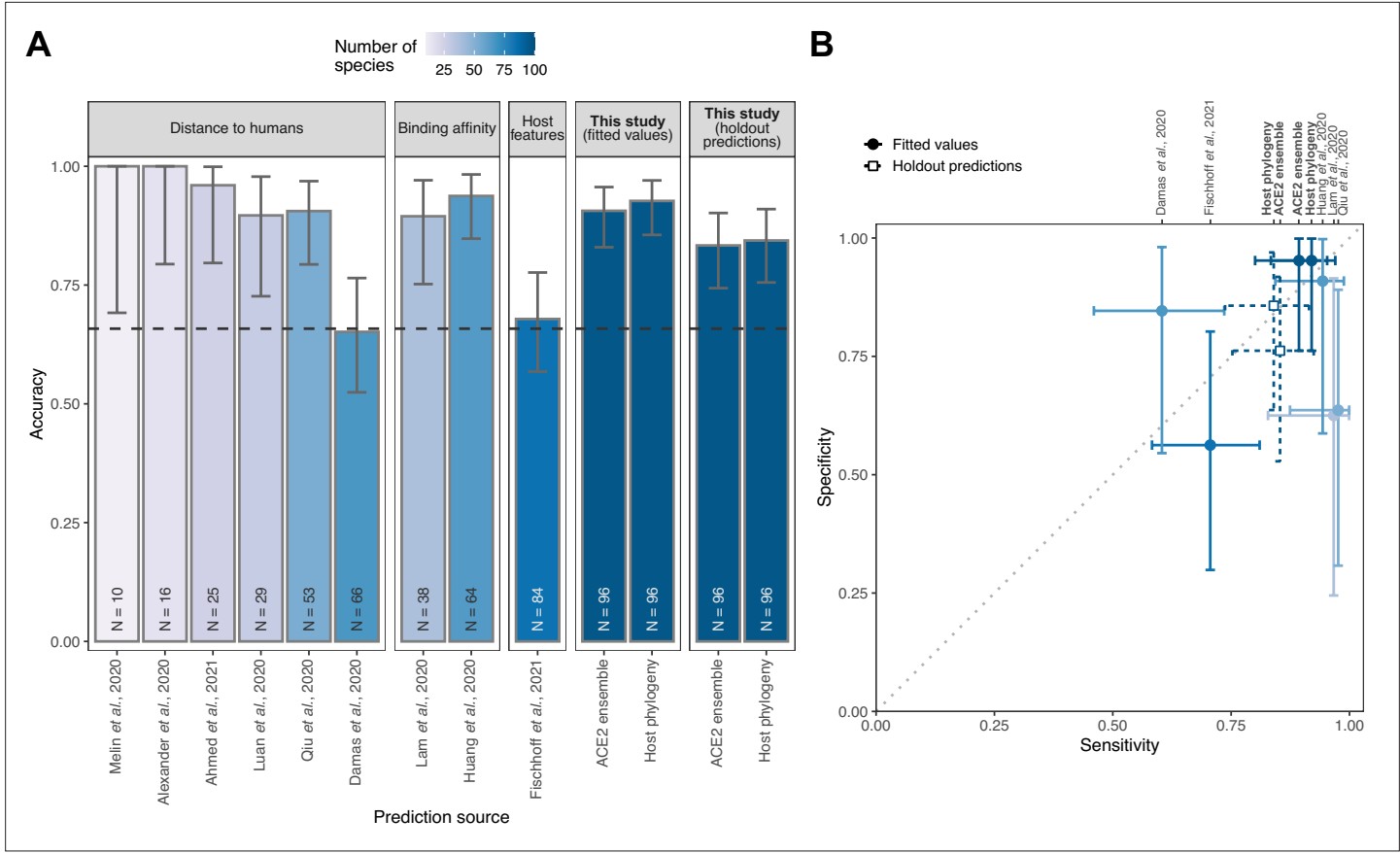

**Figure 4.** Performance of existing heuristics on our susceptibility data. (**A**) Overall accuracy, based on all species in our data for which predictions were available in each study. Accuracy measurements are arranged by increasing sample size, also indicated in colour. A dashed line indicates the performance expected from a null model which assigns labels in proportion to the frequency of susceptible species observed in the full training dataset. Note that variation in the number of species and the exact focus of predictions across studies necessitates cautious interpretation of performance metrics. (**B**) Trade-offs between sensitivity and specificity. A dotted line indicates balanced sensitivity and specificity, colours as in (**A**). For clarity, only studies providing predictions for ≥30 species in our data are shown. In both panels, error bars represent 95% binomial confidence intervals. Earlier studies provided no formal metrics of model performance and used fixed cutoff values selected based on observed susceptibility in the predicted species, rather than from independent data. This makes it impossible to calculate performance metrics directly comparable to the formal holdout-based measures used in this study. To allow comparison, we therefore include the equivalent measures of performance based on training data for our models ('fitted values'), including the best-performing ACE2-based model (AA consensus distance/AA consensus distance ensemble) and the host phylogeny-based model. Note, however, that all such measures will overestimate performance on new taxa. Performance metrics based on 'holdout predictions' (**Figure 2**) provide a more generalisable view of future performance.

The online version of this article includes the following figure supplement(s) for figure 4:

**Figure supplement 1.** Comparison of predictions across studies and models.

sites themselves. Further, a model based only on spike-binding sites performed considerably worse than other ACE2-based models (*Figure 3—figure supplement 2*).

To ensure that our machine learning approach adequately captured earlier representations of ACE2 sequences, we next compared predictions from our ACE2- and host phylogeny-based models to those of previously published heuristics. Strong correlations between predictions from our combined ACE2-based model and those from earlier studies confirmed that it was broadly representative of earlier approaches (*Figure 4—figure supplement 1*). Direct comparisons of model performance were difficult given the absence of performance metrics for previously published heuristics and wide variation in the number and identity of species for which prediction was attempted. However, performance appeared to be broadly similar across studies, with binding affinity-based predictions performing best (*Huang et al., 2020*), followed by distance-based and finally host trait-based approaches (*Figure 4*). Most distance-based approaches achieved their high accuracy by prioritising sensitivity (i.e., accurate prediction of susceptible species, which dominate current data), and had very low specificity

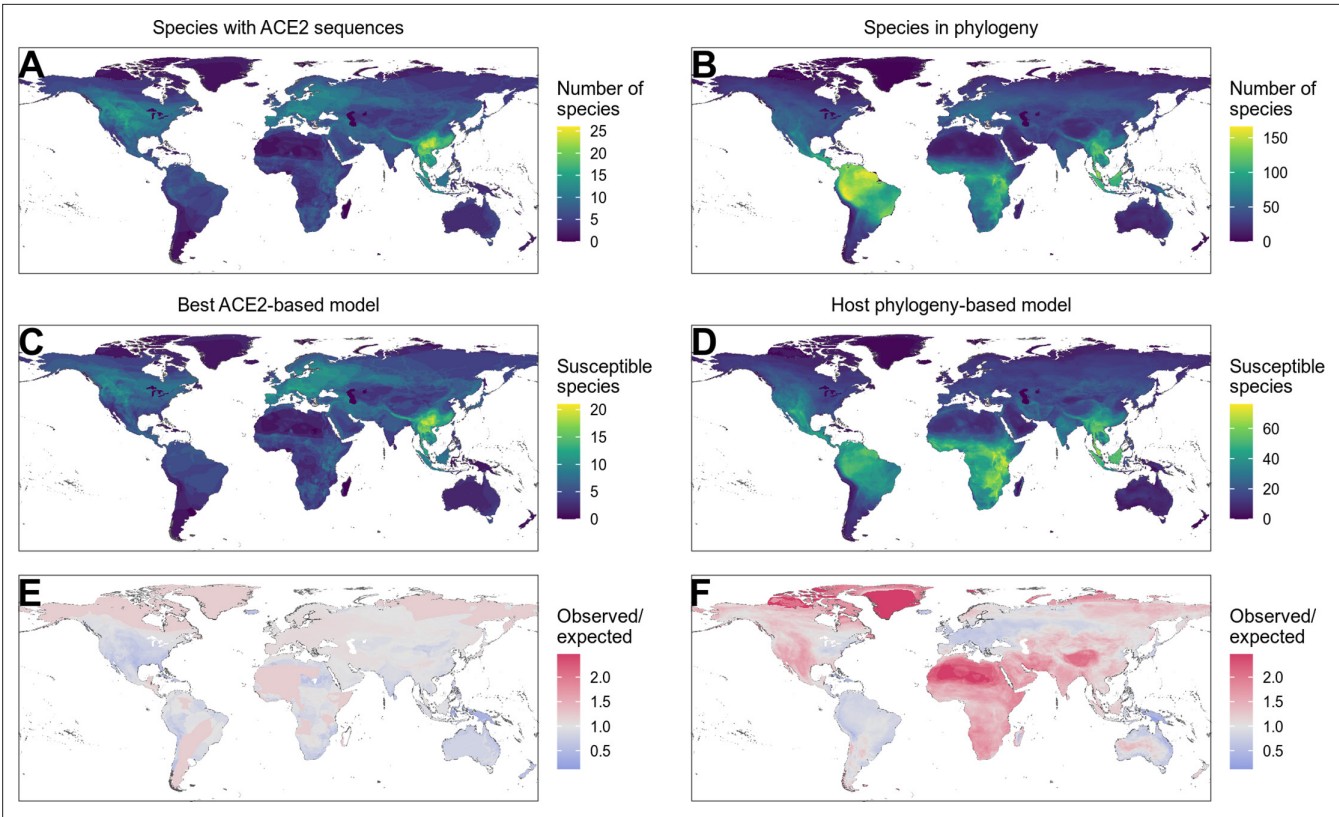

**Figure 5.** Distribution of wild terrestrial mammals predicted as susceptible depends on input data and model choice. (**A–B**) Number of species available for prediction by (**A**) ACE2-based models (limited by ACE2 availability) and (**B**) phylogeny-based models (nearly all mammals, in this figure limited primarily by the availability of IUCN range data). (**C–D**) Number of species predicted to be susceptible by (**C**) the best ACE2-based model (AA consensus distance/AA consensus distance ensemble), and (**D**) the host phylogeny-only model (phylogenetic eigenvectors). (**E–F**) Observed over expected ratio comparing the proportion of species in each location predicted as susceptible to the baseline expectation in which the overall proportion of mammal species predicted as susceptible by the relevant model is distributed homogeneously. Values below one indicate a lower proportion than expected, while values above one indicate a higher proportion than expected (i.e., putative hotspots of susceptibility). (**E**) Predictions from the best ACE2-based model; (**F**) predictions from the phylogeny-only model.

The online version of this article includes the following figure supplement(s) for figure 5:

**Figure supplement 1.** Estimating the value of quantitative susceptibility predictions from the phylogeny-only model for guiding surveillance.

**Figure supplement 2.** Species observed and predicted as susceptible to sarbecovirus infection, aggregated by taxonomic order.

**Figure supplement 3.** Proportion of species observed or predicted to be susceptible to sarbecovirus infection in boreoeutherian families.

**Figure supplement 4.** Distribution of wild terrestrial mammals predicted as susceptible by the phylogeny-only model, separated by taxonomic order.

(*Figure 4B*). The performance of our models based on fitted values (the most comparable approach to previous heuristics) was equivalent to that of the best previously published heuristic (*Huang et al., 2020*). Formal holdout-based measurements of performance for our ensemble and phylogeny-only models also overlapped with the best performing heuristics, further suggesting comparable performance, despite the more stringent evaluation. This identifies predictions based on host phylogeny as a viable alternative to ACE2 sequences for which availability remains limited.

Finally, we investigated how including host phylogeny when predicting host range would influence recommendations for targeted surveillance of sarbecovirus spillovers. We produced predictions for all mammals and birds with available ACE2 sequences or which were included in our time-scaled phylogeny, but focus our analysis on mammals since training data for birds were extremely limited (*Supplementary file 2*). Our best-performing ACE2-based model, the AA consensus distance/AA consensus distance ensemble, predicted 87.6% of mammals to be susceptible (N=178). This is broadly in line with currently available data, where 82.2% of mammals are reported to be susceptible (N=90; *Figure 1*). Mapping the distribution of terrestrial wild mammal species predicted to be susceptible by this model revealed an apparent hotspot in South-East Asia (*Figure 5C*), as previously observed by

*Fischhoff et al., 2021*. However, species in this region were not more likely to be susceptible to sarbecoviruses than expected by chance (*Figure 5E*). Instead, the observed concentration of putatively susceptible species appears to be shaped largely by the increased availability of ACE2 sequences for species from this region (*Figure 5A*). Indeed, the only regions in which more susceptible species were predicted than expected under a uniform spatial distribution were areas containing very few species with available ACE2 sequences (e.g., the arctic and the Sahara desert).

By allowing predictions on species without available ACE2 sequences, the phylogeny-only model leads to different conclusions on how surveillance might be targeted. This model reduces the proportion of susceptible mammals from 87.6% to 42.7% (N=4183), but this still implies the need for surveillance in 1785 mammal species (*Figure 5—figure supplement 1*). In contrast to the ACE2-based predictions, geographic risk was diffused across most of the tropics and subtropics with apparent hotspots limited to areas with very low species diversity (*Figure 5D and F*). Thus, this model – which performs comparably to or better than ACE2-based models – suggests that clearly defined geographic hotspots of host susceptibility suitable for targeted surveillance are unlikely to exist. Consistent with observed data, in most mammalian taxonomic orders that contained susceptible species, all included species were predicted to be susceptible (*Figure 5—figure supplement 2*). Interestingly however, primates, rodents, and bats deviated from this pattern, containing several families with no species predicted as susceptible (*Figure 5—figure supplement 3*, *Figure 5—figure supplement 4*). Thus, while the wide host range of sarbecoviruses makes it impractical to specifically target surveillance, it may be possible to exclude some groups from consideration.

## Discussion

Frequent spillovers of SARS-CoV-2 from human to non-human hosts has led to a surge in the development of heuristics that seek to predict which species are likely to be susceptible. However, the predictive power of these heuristics or their utility for guiding research and surveillance remained unclear. By cataloguing available data on sarbecovirus host range, we show that while a variety of ACE2-based approaches produce relatively sensitive and specific predictions, these predictions largely derive from strong correlations with host phylogeny, and limitations and biases in their input data limit their actionability. Models based on host phylogeny alone perform equivalently, enabling scalable prediction across nearly all mammals, but imply that in the absence of additional metrics of inter-species exposures, vast numbers of species would need to be surveyed with limited geographic or taxonomic focus.

ACE2 sequences showed surprising accuracy for predicting sarbecovirus host range, both in the formally trained models produced here and in earlier heuristics. However, various lines of evidence suggest the predictive power of ACE2-based models derives primarily from the correlation of ACE2 variation with host phylogeny. First, susceptibility to sarbecovirus infection was highly conserved, clustering in patterns consistent with the evolutionary history of potential host species. Second, ACE2 is also evolutionarily conserved, and a phylogeny derived from ACE2 sequences was highly congruent with one reflecting broader evolutionary history. Third, commonly used approaches for representing ACE2 sequence differences are largely equivalent in the amount of susceptibility information carried (*Figure 2*, *Figure 4—figure supplement 1*), despite measuring biologically distinct aspects of ACE2 sequence variation and its interaction with sarbecovirus spike proteins. Fourth, models based on host evolutionary history performed similarly to the best ACE2-based models, while considering ACE2 in addition to evolutionary history provided no detectable improvements. Finally, in ACE2-based models with access to information about individual amino acid positions, most predictive power was derived from phylogenetically-informative sites rather than virus-binding sites, even when measures of binding affinity and sequence similarity were removed. This held true for both rhinolophid bats (the putative ancestral hosts of sarbecoviruses which may have different host-virus interactions due to long-term co-evolutionary dynamics) and other species, with the same features and sites used across all species without affecting predictive accuracy (*Figure 3—figure supplement 1*, *Figure 3—figure supplement 3*). Taken together, these results question the exclusive focus on ACE2 as a predictor of sarbecovirus host range. Biologically, the apparent success of ACE2-based models in predicting sarbecovirus host range cannot be interpreted as evidence for a disproportionate mechanistic influence of ACE2 variation in restricting host range. While there is experimental evidence suggesting that variation in ACE2 alone can block infection (e.g., *Guo et al., 2020*; *Thakur et al., 2022*), how widely alleles conferring

complete resistance occur in nature (particularly in species with no history of sarbecovirus exposure) remains unclear. Indeed, our data show that a broad diversity of ACE2 variation is permissive for infection, so such examples, though biologically verified, may be extremes that do not generalise across taxa. Instead, sarbecovirus host range is likely to be shaped by a range of factors all showing some level of phylogenetic conservation (e.g., receptor distribution, innate immunity, behaviour). These results also mean caution is needed when interpreting the predictive power of specific sites in ACE2 sequences, as phylogenetic correlation means their biological significance cannot be adequately assessed from observational data.

A key drawback of ACE2-based models is that data are only available for a small fraction of potential host species (*Figure 5A*). This limits their applied use and may also introduce biases since many ACE2 sequences likely stem from existing sarbecovirus research which is geographically biased. One previous effort circumvented the limited availability of ACE2 sequences using host traits believed to follow similar patterns of phylogenetic conservation as ACE2 sequences (*Fischhoff et al., 2021*). However, although we reiterate that quantitative measures of performance should be considered cautiously, this approach appeared to have reduced performance relative to models directly incorporating host phylogeny (*Figure 4*). Further, by conditioning a key stage of model development on ACE2 binding, we speculate that this approach may have introduced the same biases plaguing ACE2 sequences. For example, the top 10% of hosts presented as capable of transmitting SARS-CoV-2 by *Fischhoff et al., 2021*, have a similar geographic distribution to our ACE2-based models (*Figure 5C*), which in turn mirror overall geographic biases in mammalian ACE2 sequence availability (*Figure 5A*). Our results show host phylogeny as a viable alternative, which performs comparably to the best ACE2-based models. Anecdotally, this accuracy extends to species only recognised as susceptible after data collection for this study had ended, with 11 of 12 species more recently reported as naturally infected accurately predicted by the phylogeny-based model (*Supplementary file 2*; *Allender et al., 2022*; *Kuchipudi et al., 2022*; *Padilla-Blanco et al., 2022*; *Pereira et al., 2022*; *United States Department of Agriculture, 2022*; *United States Department of Agriculture, 2021a*; *United States Department of Agriculture, 2021b*; *United States Department of Agriculture, 2021c*; *Wang et al., 2022*). However, beyond data availability, the wide host range of sarbecoviruses represents a more significant practical hurdle, since the scope of surveillance suggested by these models is massive. This issue does not stem from the models, which show high accuracy on current data, but rather are representative of a pattern of widespread susceptibility seen consistently across experimental infections (65.2% of mammals susceptible, CI: 42.7–83.6%), our wider training data (82.2%, CI: 72.7–89.5%), and predictions from both the ACE2-based ensemble and host phylogeny-based models (87.6% and 42.6%, respectively). In essence, the potential host range of sarbecoviruses may be genuinely too broad for accurate predictions to be actionable in the absence of information on the likelihood of cross-species exposures.

Given the broad predicted host range of sarbecoviruses, how should research and surveillance actions proceed? While it may be possible to improve predictive performance further by considering variation in additional host proteins linked to infection (e.g., TMPRSS2; *Hoffmann et al., 2020*), this is unlikely to be practically useful given expected correlation with phylogeny and the low numbers of species for which sequences of multiple relevant genes will be realistically attainable. Such an approach is also unlikely to significantly reduce the number of susceptible hosts predicted. Alternatively, surveillance might be better informed by targeting hosts in proportion to their predicted risk. However, while conditioning our predictions on the top 30% of species captures the majority of currently known susceptibles (73%), this still means at least 1255 mammals require surveillance while introducing a significant risk of missing key spillovers (*Figure 5—figure supplement 1*). In the interim, surveillance might therefore target species for which local knowledge indicates high human-animal contact, further refined by our model predictions (e.g., knowledge of likely non-susceptible groups, *Figure 5—figure supplement 2*, *Figure 5—figure supplement 3*). Looking further into the future, our data suggest that the majority of susceptible hosts are unlikely to transmit the virus effectively (*Figure 1A*), and indeed only a limited number of natural spillovers have thus far resulted in onward transmission (*Kuchipudi et al., 2022*; *Oude Munnink et al., 2021*). Predicting broader reservoir competence instead of susceptibility to infection may therefore be a path forward to optimise surveillance (*Becker et al., 2020*). However, the inability of current data to produce models capable of accurately predicting virus shedding (*Figure 2—figure*

*supplement 1*) indicates a need for greater investment to gather sufficient data on the ability of hosts to transmit sarbecoviruses. Careful interrogation of observed spillover events to evaluate sustained transmission (*Kuchipudi et al., 2022*; *Oude Munnink et al., 2021*) and inclusion of naïve hosts in experimental infections to track transmission (*Freuling et al., 2020*; *Shi et al., 2020*) would be particularly valuable.

A clear limitation of our approach is the relatively small number of species for which sarbecovirus susceptibility data are available. Combining diverse data sources and expanding the scope to all sarbecoviruses allowed us to evaluate the rapidly growing number of untested heuristics predicting SARS-CoV-2 host range. However, limited numbers of known non-susceptible species in particular made it difficult to distinguish variation in the specificity of alternative approaches with confidence, a problem exacerbated by low levels of overlap between available data and the species for which predictions were available in individual studies (wide CIs, *Figure 4B*). Although our strategy to maximise sample size meant including data from cell culture-based experiments, we found no evidence to suggest that this skewed results (*Figure 2—figure supplement 2*). Nevertheless, once more data are available it may be more appropriate to train separate models for each data type (i.e., separate models predicting receptor binding, susceptibility in experimental infections, etc.), as this would allow training an ensemble model to learn the optimal weighting of predictions based on different data types when predicting real-world susceptibility. We were also unable to incorporate polymorphism in ACE2 sequences or suspected variation in susceptibility below the species level, which has been reported in some rhinolophid bats (*Guo et al., 2020*), since these fine-scale data were unavailable for most species. SARS-CoV-2 is also continuously evolving and this evolution may ultimately alter host range in ways that cannot be predicted by current models (*Thakur et al., 2022*), which are necessarily based on infections reported with earlier strains. While previous approaches were not explicitly designed for non-SARS-CoV-2 sarbecoviruses, our results suggest the inclusion of additional sarbecoviruses did not adversely impact model performance (*Figure 2—figure supplement 3*, *Figure 2—figure supplement 4*, *Figure 4—figure supplement 1*). This may be because most species (90.8%) included in our dataset have been either linked to or tested for susceptibility to sarbecoviruses known to use ACE2 as an entry receptor (*Khaledian et al., 2022*; *Starr et al., 2022*), including all non-susceptible species (*Supplementary file 1*). Nine susceptible host species have so far only been reported to be infected by sarbecoviruses which are either not known to use ACE2 (due to a lack of published experiments involving the viruses in question) or are suspected to use an alternative receptor (based on an inability to bind the ACE2 orthologs tested thus far); we detected no differences in accuracy when predicting the susceptibility of these species (*Figure 2—figure supplement 3*, *Figure 2—figure supplement 4*, *Figure 4—figure supplement 1*). This convergence in performance for viruses that putatively use different receptors lends further support to our conclusion that sarbecovirus host range is determined by factors other than ACE2 sequence variation. Finally, available susceptibility data may be affected by the same taxonomic and geographic biases affecting ACE2 sequence availability. For example, directly observed data will be biased to countries with capacity for surveillance, in vivo data will be limited to animal models that can be maintained in captivity, and heterologous ACE2 experiments are limited by requiring prior knowledge of ACE2 sequences. However, while this implies that predictions could be periodically updated as new observations become available, we see no mechanism through which current biases could produce a false equivalence between ACE2 sequences and host phylogeny.

Our results imply major challenges for predicting the host range of generalist viruses such as SARS-CoV-2, even when models are reasonably accurate. Both current data and model predictions suggest a significant ongoing risk of reverse zoonosis to a wide range of species, implying a need for continued surveillance among species in contact with humans. Our results provide quantitatively validated predictions of sarbecovirus susceptibility across all mammals, which together with complementary data on exposure risk could be used to narrow such surveillance. However, they also highlight a data gap which currently precludes models that predict not just susceptibility but competence for onward transmission. Such models will be vital to understand the risk that reverse zoonosis poses for establishing novel reservoirs of SARS-CoV-2 genetic diversity that may compromise global efforts to control the ongoing pandemic.

## Methods

### Data collection

Data on sarbecovirus host range were collected through literature searches and targeted follow-up of news reports, focusing on identifying reports of natural spillover and experimental infections. These were supplemented with records from cell culture infection assays and two studies which used functional assays in which cells expressing heterologous ACE2 from various species were inoculated with SARS-CoV-2 (*Liu et al., 2021*; *Mykytyn et al., 2021*). This allowed us to test the utility of adding in vitro data when information on susceptibility is limited at the organismal level. In all cases, data were treated in a hierarchical manner, preferentially recording susceptibility from the most reliable data type found (natural infection > experimental infection > cell culture > heterologous ACE2 experiment). Data were summarised to the host species level, allowing more reliable data sources to overrule observations from other sources (e.g., an observed natural infection was allowed to overrule negative experimental infections involving the same species, as such experiments may have failed for a variety of reasons such as insufficient dose). Where multiple equivalent studies provided conflicting evidence for the same species (e.g., one successful and one unsuccessful experimental infection), we recorded the species as susceptible, reflecting our desire to predict whether a species had ever been observed as susceptible to any sarbecovirus.

We also recorded whether or not species reported as susceptible were capable of shedding infectious virus, either through reports of onward transmission (e.g., to co-housed naïve individuals) or through isolation of virus from nasal or rectal swabs or from faeces. Species were recorded as negative for virus excretion only when either virus isolation or co-housing with naïve individuals was specifically attempted. These data were available for 23 species.

Sequence data for ACE2 orthologs were obtained from the NCBI Gene database, retaining the first reference sequence accession number listed for each species (GeneID: 59272; list of all orthologs downloaded 16 March 2022). This provided us with a curated set of sequences – maximising data quality – while also avoiding any biases which may be introduced when attempting to incorporate multiple ACE2 sequences for the limited number of species for which more than one sequence was available. This NCBI-curated dataset was supplemented by specific searches for host species with available susceptibility data to identify additional ACE2 sequences. For species with susceptibility data but no available ACE2 sequence, we used ACE2 sequences from their closest available relative (based on a time-scaled phylogeny from TimeTree; *Kumar et al., 2017*). This replacement affected 21 species in the final dataset, and most (62%) involved replacement sequences from the same genus (*Supplementary file 1*).

### Evaluating phylogenetic clustering

Amino acid sequences for all ACE2 orthologs were downloaded from the NCBI protein database and aligned using the E-INS-i option of MAFFT version 7.471 (*Katoh and Standley, 2013*). A maximum likelihood phylogeny was then generated using version 1.6.12 of IQ-TREE (*Minh et al., 2020*), with built-in model selection used to choose the substitution model minimising the Bayesian information criterion (*Kalyaanamoorthy et al., 2017*). The best model used the Jones, Taylor, and Thornton substitution matrix (*Jones et al., 1992*) with empirical amino acid frequencies and six categories of rate variation in the 'FreeRate' model of *Soubrier et al., 2012*.

We also obtained a phylogeny reflecting the estimated divergence dates of all amniotes from TimeTree version 4 (*Kumar et al., 2017*). To allow comparison between phylogenies, the time-scaled phylogeny was trimmed to species with available ACE2 sequences, before calculating Baker's $\gamma$ correlation coefficient (*Baker, 1974*) using version 1.15.1 of the dendextend library in R version 4.1.0 (*Galili, 2015*; *R Development Core Team, 2021*) along with version 2.1.0 of the phylogram library (*Wilkinson and Davy, 2018*). To calculate the expected null distribution of $\gamma$ values given these particular phylogenies, species names on both phylogenies were randomly shuffled 1000 times.

To measure the amount of phylogenetic clustering among susceptible and non-susceptible species, we calculated Pagel's $\lambda$ using version 0.7_80 of the phytools R library (*Pagel, 1999*; *Revell, 2012*). Pagel's $\lambda$ was re-calculated while cumulatively adding data from species with susceptibility or non-susceptibility known from natural infection, experimental infection, cell culture, and finally heterologous ACE2 experiments. We also compared distances to the closest other susceptible or non-susceptible species, using both cophenetic distances and ACE2 amino acid distances. Cophenetic distances were

calculated on the time-scaled (TimeTree) phylogeny using version 5.5 of the APE R library (*Paradis et al., 2004*), while amino acid distances were calculated as the total Grantham distance separating any two species across all sites in the ACE2 alignment (*Grantham, 1974*).

## Sequence representations

Current heuristics for predicting susceptibility to SARS-CoV-2 infection focus on either some measure of ACE2 amino acid distance from its human counterpart or the predicted binding affinity between each ACE2 protein and the SARS-CoV-2 spike protein. To compare such metrics in a common framework, we calculated or obtained a range of amino acid sequence representations that broadly captured current approaches alongside novel alternatives. This included binding affinities from two recent studies (*Fischhoff et al., 2021*; *Huang et al., 2020*) ('binding affinity', represented as 2 separate feature columns). Neither study supplied binding affinity measures for all ACE2 sequences in our training data, but combined ensured at least one value for each included sequence. To represent the widely used distance to human ACE2, we calculated the total amino acid distance between each species' ACE2 protein and that of humans (termed 'distance to humans' in all figures, and calculated as described above, 1 feature). Despite variation in which amino acid positions were included by previous distance-based studies, all measures were highly correlated (*Figure 4—figure supplement 1*), making the total amino acid distance (i.e., the distance calculated while considering all amino acid positions) to human ACE2 a reasonable proxy. Although this metric is commonly used, whether human ACE2 is the appropriate baseline is unclear given the relatively recent association between SARS-CoV-2 and humans. We therefore also included a similar metric capturing the total amino acid distance to the ACE2 of the closest known susceptible species. This distance was calculated with reference to the relevant training dataset only, and separately expressed the distance to the closest susceptible known from either direct observations, experimental infections, heterologous ACE2 experiments, or across all data ('distance to susceptible', 4 features; data contained no susceptibles known only from cell culture experiments).

We also sought to include more nuanced information about specific ACE2 sites by including representations of individual positions in the ACE2 protein alignment. The amino acids observed at each alignment position showing variation was represented using one-hot encoding of all amino acids observed at a frequency >10%, alongside an additional binary variable taking a value of 1 for all amino acids observed less frequently at a given position, and 0 otherwise ('AA categorical', 1657 features). We also summarised the amino acids at each variable alignment position using the physicochemical properties hydrophobicity, polarity, net charge, and Van der Waals volume ('AA properties', 1841 features), with values for individual amino acids obtained from the AAindex database (accessions JURD980101, ZIMJ680103, KLEP840101, FAUJ880103) (*Fauchère et al., 1988*; *Juretić et al., 1998*; *Kawashima et al., 1999*; *Klein et al., 1984*; *Zimmerman et al., 1968*). Finally, we included the Grantham distance between each observed amino acid and the most common amino acid observed at that site among susceptible species ('AA consensus distance', calculated with reference to the relevant training dataset only, 531 features).

## Phylogenetic eigenvectors

The machine learning algorithms we used (described below) cannot directly represent phylogenetic information. Therefore, to include evolutionary relationships among possible hosts in our models, we used phylogenetic eigenvectors calculated from the time-scaled phylogeny described above and including all amniotes. Phylogenetic eigenvectors were calculated using version 0.3–6 of the MPSEM library in R, assuming Brownian motion (*Guénard et al., 2013*). The first 50 eigenvectors were retained and included in models ('phylogenetic eigenvectors', 50 features).

## Machine learning

We trained a series of gradient-boosted classification tree models to predict the susceptibility of individual species using various combinations of ACE2 sequence representations or the phylogenetic eigenvectors. Models were constructed using XGBoost version 1.4.0 in R alongside version 0.1.3 of the tidymodels suite of helper libraries (*Chen and Guestrin, 2016*; *Kuhn and Wickham, 2020*). Given the relatively small amount of training data available, performance was evaluated using leave-one-out

cross-validation. Throughout, we report proportions of species accurately predicted, alongside binomial CIs calculated using the binom.test function in R (*Clopper and Pearson, 1934*).

Models were trained in a nested process to allow simultaneous tuning and evaluation of performance. For each species, data were split into training data and the single holdout species for cross-validation ('outer split'). The training data were then randomly split again, keeping 75% for model training with a given combination of tuning parameters, with performance evaluated on the remaining data ('inner split'). This inner split was repeated for each unique tuning parameter combination, testing 100 parameter combinations across a maximum entropy grid (*Dupuy et al., 2015*). After training an initial model on a given tuning parameter combination, a search was performed for the cutoff which best balanced sensitivity and specificity on the inner split's testing data when converting quantitative scores produced by the model into binary predictions. This was achieved by calculating the mean of sensitivity and specificity ('balanced accuracy') for each cutoff. The tuning parameter and cutoff combination which maximised balanced accuracy was used to train a final model on all training data (i.e., all data except for the single species held back in the outer split). This nested tuning and evaluation process was repeated to generate holdout predictions for all species, allowing calculation of the final reported performance statistics.

## Ensemble models
In addition to the direct combination of different ACE2 sequence representations and phylogenetic eigenvectors in the same model, we also tested whether training separate models before combining predictions would improve performance. Models were combined by averaging the quantitative holdout predictions for each species. To convert these averages to binary predictions, we calculated the optimal cutoff balancing sensitivity and specificity as described above, using the averaged quantitative predictions for all species except a given focal species to evaluate potential cutoffs, before using the best cutoff to generate a binary prediction for the focal species. This procedure was repeated to generate predictions for all species.

## Feature importance
The importance of individual features in shaping model predictions was measured using SHAP values (*Lundberg and Lee, 2017*), which measure the contribution of individual features to the final quantitative prediction for a given host species. Using a final model trained on all available data, we calculated feature importance as the mean of absolute SHAP values across all species.

Since many features represented individual ACE2 positions which are expected to co-vary, we also sought to characterise the correlation between positions. For each position in a given ACE2 sequence, we first calculated the Grantham distance between the observed amino acid and the most common amino acid observed at that position across all sequences used in model training. We then calculated Spearman correlations in this distance metric for all possible pairs of ACE2 positions, before using the correlations to perform affinity propagation clustering with version 1.4.8 of the apcluster library in R (*Bodenhofer et al., 2011*; *Frey and Dueck, 2007*). The resulting clusters were used to annotate positions in *Figure 3*, separating known spike-binding sites ('S-binding'), sites occurring in the same cluster as a known spike-binding site ('correlated with S-binding site'), and all other sites ('other').

## Comparison of other studies
To compare predictions across our models and to those from previously published heuristics, we collected both binary predictions and the underlying quantitative scores/heuristics used to make these predictions from 12 studies. We compared predictions at the species level. In rare cases where previous studies provided predictions for multiple representatives of a given species (e.g., predictions for subspecies or for different ACE2 sequences of the same species), we used the mean of quantitative scores and the most common binary prediction. Scores from each study were re-scaled to lie between 0 and 1, allowing us to plot them on the same scale. Where needed, scores were reversed to ensure that higher values always reflected species considered more likely to be susceptible.

The similarity of predictions across our models and earlier heuristics was summarised by calculating pairwise Spearman correlations, using all overlapping species in each pair of models/heuristics. These correlations were used for agglomerative hierarchical clustering, using the mean

correlation across all points as the distance metric between clusters (i.e., the unweighted pair-group arithmetic average method) in version 2.1.2 of the cluster R library (*Kaufman and Rousseeuw, 1990*).

For studies which supplied binary or categorical predictions, we also estimated sensitivity, specificity, and overall accuracy, all in light of the new susceptibility dataset generated here. Two studies supplied categorical rather than binary (susceptible/non-susceptible) predictions. In the case of *Melin et al., 2020*, we treated species predicted as having high susceptibility as susceptible, and those predicted to have low susceptibility as non-susceptible. Similarly, species predicted as having very high, high, or medium susceptibility by *Damas et al., 2020*, were considered susceptible, treating predictions of low and very low susceptibility as non-susceptible. A third study, *Fischhoff et al., 2021*, transformed observed ACE2-sarbecovirus binding strength values into binary labels where the 'positive' category was believed to reflect capacity for both infection and virus shedding and trained models to predict this category for other species. Since this prediction category was analogous to other studies (i.e., using a fixed threshold of binding strength as a proxy for the infection variable of interest), we included them in our model comparisons, treating all species with positive predictions as susceptible.

Importantly, performance estimates for all published heuristics relied on predictions as published. Since these heuristics did not withhold species with known infection outcomes as an independent test set, cutoffs chosen to judge susceptibility or non-susceptibility to SARS-CoV-2 would have been defined by some or all species to which the heuristic was applied (e.g., selecting minimum heuristic values which appeared consistent with infection records). This is likely to artificially inflate their performance beyond what could be expected when applying these heuristics to host species that were not involved in the development of the heuristic. In contrast, our approach of withholding empirical susceptibility records during model development gives a more accurate representation of how performance will generalise to new potential host species.

## Mapping predictions

To visualise the spatial distribution of species predicted as susceptible (and of available ACE2 sequences), we summarised IUCN range maps for terrestrial mammals into a grid with cells measuring $1/6 \times 1/6$ of a degree. Using predictions from models trained on all available data, we counted the number of susceptible species with ranges overlapping each grid cell. We also measured whether the proportion of predicted susceptible species in any one grid cell was higher than expected by chance by dividing the proportion of species predicted as susceptible in each cell by the overall proportion of species predicted as susceptible across the entire dataset. This meant we were comparing the observed frequency of susceptible species in any one location to a baseline assumption of uniform geographic distribution of susceptible species.

## Acknowledgements

We thank members of the Streicker lab for feedback on earlier versions of this manuscript. NM and DGS were supported by a Wellcome Senior Research Fellowship (217221/Z/19/Z) and the Medical Research Council (MC_UU_12014/12). The funders had no role in study design, data collection, and interpretation, or the decision to submit the work for publication.

## Additional information

### Funding

| Funder | Grant reference number | Author |
| --- | --- | --- |
| Medical Research Council | MC_UU_12014/12 | Nardus Mollentze<br>Daniel G Streicker |
| Wellcome Trust | 217221/Z/19/Z | Nardus Mollentze<br>Daniel G Streicker |

| Funder | Grant reference number | Author |
|---|---|---|

The funders had no role in study design, data collection and interpretation, or the decision to submit the work for publication. For the purpose of Open Access, the authors have applied a CC BY public copyright license to any Author Accepted Manuscript version arising from this submission.

## Author contributions

Nardus Mollentze, Conceptualization, Data curation, Formal analysis, Investigation, Visualization, Writing - original draft, Writing - review and editing; Deborah Keen, Data curation, Investigation; Uuriintuya Munkhbayar, Data curation; Roman Biek, Conceptualization, Supervision, Investigation, Writing - review and editing; Daniel G Streicker, Conceptualization, Supervision, Funding acquisition, Investigation, Writing - review and editing

## Author ORCIDs

Nardus Mollentze http://orcid.org/0000-0002-2452-6416
Deborah Keen http://orcid.org/0000-0002-6735-6330
Roman Biek http://orcid.org/0000-0003-3471-5357
Daniel G Streicker http://orcid.org/0000-0001-7475-2705

## Decision letter and Author response

Decision letter https://doi.org/10.7554/eLife.80329.sa1
Author response https://doi.org/10.7554/eLife.80329.sa2

## Additional files

### Supplementary files

• Supplementary file 1. Final data on susceptibility to, and shedding of, sarbecoviruses, along with the accession numbers for angiotensin-converting enzyme 2 (ACE2) amino acid sequences used to represent each species. These data were used to train all models presented.

• Supplementary file 2. Predictions from the phylogeny-only model. Predictions are separated into four categories, reflecting predictions for (a) species in the training data, produced by withholding each species in turn from model training (i.e., predictions informing the leave-one-out cross-validation statistics presented), (b) susceptible species recognised through natural infection after data collection for this study had ended, (c) all other mammals available in the TimeTree phylogeny, and (d) all other birds in the TimeTree phylogeny. Across all tables, 'cutoff' refers to the optimised value beyond which predicted probabilities from a given model are considered to indicate susceptible species (labelled as 'True' in the 'prediction' column).

• MDAR checklist

### Data availability

Data and code required to reproduce all analyses are available at https://doi.org/10.5281/zenodo. 6552861.

The following dataset was generated:

| Author(s) | Year | Dataset title | Dataset URL | Database and Identifier |
|---|---|---|---|---|
| Mollentze N, Keen D, Munkhbayar U, Biek R, Streicker DG | 2022 | Variation in the ACE2 receptor has limited utility for SARS-CoV-2 host prediction | https://doi.org/10.5281/zenodo.6552861 | Zenodo, 10.5281/zenodo.6552861 |

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
