## [Editor Report]

Many approaches to predict which animals species might be at risk of infection by SARS-CoV-2 focus on features of the ACE2 host cell receptor to which the virus binds. This important study shows that such methods are not uncovering a true biological signal. Instead, Mollentze and colleagues show that ACE2 sequences are effectively only a proxy for generic species relationships, and species phylogeny alone can provide equivalent predictive power.

---

## [Decision Letter]

**Decision letter after peer review:**

Thank you for submitting your article "Variation in the ACE2 receptor has limited utility for SARS-CoV-2 host prediction" for consideration by *eLife*. I apologize for the delay in the review process.

Your article has been reviewed by 3 peer reviewers, and the evaluation has been overseen by a Reviewing Editor and Miles Davenport as the Senior Editor. The following individual involved in review of your submission has agreed to reveal their identity: Tyler Starr (Reviewer #1).

Essential revisions:

We have decided there are no truly essential revisions. However, the reviewers have suggested ways to strengthen the writing and analysis. Please consider them carefully. We deem the requested clarifications important enough to merit a revision or reply.

*Reviewer #1 (Recommendations for the authors):*

1. I found the logical flow a bit indirect. As the authors point out in the introduction, ACE2 binding is necessary but not sufficient in conferring susceptibility. However, the focus they then ascribe to ACE2-based data and ACE2-based predictions weakens the forcefulness with which they introduce this idea that determinants beyond ACE2 binding are going to be important in a full evaluation of animal susceptibilities. For example, the plurality of points in their collated dataset on "susceptibility" derives from (although acknowledges and de-prioritizes the significance of) heterologous ACE2 expression in cell culture viral entry assays, thereby missing all downstream determinants of susceptibility. Furthermore, more and more elaborate models based on ACE2 alone are built (e.g. amino acid classifications, site-wise determinants, ACE2 distance). My interpretation is that this is an attempt to "do the best that can be done" with ACE2 sequence alone so as not to create a straw-man argument for the ACE2 sequence features to then compare to phylogeny alone. This is why I present this as a logical flow issue, and perhaps not a scientific issue. Some ideas to improve this flow could be to spend more of the Introduction emphasizing the importance the field typically puts on ACE2 sequence alone and not as forcefully explain why this is unlikely to be a sufficient proxy (can leave this description until results or discussion), leaving the punchiness of the conclusion that it is indeed not that powerful as a less obvious outcome than is currently presented after the Introduction.

2. Toward the question of the sufficiency of ACE2 binding data alone for predicting susceptibility: when evaluating the hierarchy of evidence used in the final collated dataset (where animal-based studies trump heterologous ACE2 cell culture experiments), were there any cases where an ACE2 receptor is known to be sufficient to enable cell entry but the animal itself is not experimentally susceptible? Highlighting any observations of this seems the most direct evidence to the point that ACE2 sequence alone is insufficient to predict susceptibility.

3. I wonder whether a more pseudo-mechanistic two-step model could be considered or proposed in the Discussion – a first step based on ACE2 sequence (the "first step" necessary for susceptibility), and the second based on broader determinants of susceptibility (which would be best captured by phylogeny given complexity of unknown unknowns). The reason this may be helpful is that ACE2 binding can be "flipped" on or off with individual amino acid mutations and therefore more quickly deviate from phylogenetic trends (e.g. due to virus-host arms races in Rhinolophus bats, PMID 32699095, but see point 6 below). The broader determinants of susceptibility involve many complex components from cell biology to physiology, and so of course can not be tractably ascribed to a single gene sequence, but also may be better captured by a phylogenetic scale anyway. Such a two-tiered model could also better accommodate the multi-tiered data that is collapsed into a single training paradigm in the current study – for example, the simpler heterologous ACE2 entry assays could be incorporated only for the "first step" of model evaluation, while the whole-animal susceptibility data could serve as a target for the second step / integrated two-step model. This model would lose the utility of the phylogenetic model as illustrated in Figure 5, in that ACE2 sequence would need to be known, but could better incorporate multi-modal data to improve predictive accuracy.

4. Many species of bats show dramatic ACE2 polymorphism centered on positions contacted by sarbecoviruses, and this variation is known to influence binding of certain sarbecoviruses (e.g., PMID 32699095). How did you account for ACE2 polymorphism in analyses -- did you just resolve each species to a single ACE2 sequence? And how do you incorporate into the model possibilities that certain ACE2 alleles within a single species are permissive to entry by some sarbecoviruses while others are not?

5. Related to the above point (and a concern that I had about species like R. pearsonii which are seemingly not susceptible to ACE2-utilizing sarbecoviruses, but then I saw Figure 2 – supplement 2): given that the dynamics of long-term host:virus coevolution that exist in Rhinolophus bats are so different from the dynamics of susceptibility that are germane to questions of reverse zoonotic and intermediate/amplifying potential of other species – might it be wise to exclude Rhinolophus species from the analyses as they may require different forms of "signal" for prediction of susceptibility compared to the rest of mammals?

*Reviewer #2 (Recommendations for the authors):*

Pg 2 L11-12 – has anyone looked at the correlation between susceptibility in cell culture vs in vivo (for any virus)? Might be complex as often comparing across tissue types as well as species. Some support from https://doi.org/10.1371/journal.ppat.1004475 but not sure if there are other studies such as this looking directly at susceptibility?

Pg 3 – L23-30 – re the assumption that infection with SARS-CoV suggests susceptibility also to SARS-CoV-2 and vice versa – this seems reasonable, although may expect some instances of virus by species interactions meaning this is not the case.. Looking at the supp data it looks like in the 6 species tested with both SARS^-1^ and -2 they can be infected by both (ie 6/6) – is that correct? If so maybe alter the text to say this specifically, and offer some more general support for your approach with (eg their own recent paper in PNAS or https://onlinelibrary.wiley.com/doi/10.1111/tbed.14361)

Pg 4 L3-10 felt some more detail was needed here on ACE2 data to help the reader follow the approach

The authors use the described modelling approach to predict what species are susceptible to sarbecovirus infection, aggregated by taxonomic order (Figure 5, Figure Supplement 2). From the figure it is apparent that the observed data set contains primarily mammalian samples where as the predicted dataset contains a considerably higher proportion of avian samples. Given that the narrative around reverse zoonosis is largely focused on mammals, it would be helpful to have more of a discussion around the role of avian species both in these analyses and in the transmission of sarbecoviruses.

Data and scripts are available via a doi.

*Reviewer #3 (Recommendations for the authors):*

You write

"Second, since no model has been trained or validated using observed data on infection (the outcome of interest)"

Then later:

"Understanding the value of ACE2-based host range predictions for guiding surveillance therefore requires developing models based on the outcome of interest – susceptibility – and quantifying the accuracy of their predictions."

And later again:

"We treated these different sources of information hierarchically, considering the best available evidence for compatibility or incompatibility in each host species (natural infection > experimental infection > cell culture > heterologous ACE2 cell culture experiments)."

I'd recommend changing "infection" to susceptibility in the first-quoted passage. To me (maybe just me?) "infection" makes me think infection of a live animal, even though I know that cells in culture can be infected too.

---

## [Author Response]

Reviewer #1 (Recommendations for the authors):1. I found the logical flow a bit indirect. As the authors point out in the introduction, ACE2 binding is necessary but not sufficient in conferring susceptibility. However, the focus they then ascribe to ACE2-based data and ACE2-based predictions weakens the forcefulness with which they introduce this idea that determinants beyond ACE2 binding are going to be important in a full evaluation of animal susceptibilities. For example, the plurality of points in their collated dataset on "susceptibility" derives from (although acknowledges and de-prioritizes the significance of) heterologous ACE2 expression in cell culture viral entry assays, thereby missing all downstream determinants of susceptibility. Furthermore, more and more elaborate models based on ACE2 alone are built (e.g. amino acid classifications, site-wise determinants, ACE2 distance). My interpretation is that this is an attempt to "do the best that can be done" with ACE2 sequence alone so as not to create a straw-man argument for the ACE2 sequence features to then compare to phylogeny alone. This is why I present this as a logical flow issue, and perhaps not a scientific issue. Some ideas to improve this flow could be to spend more of the Introduction emphasizing the importance the field typically puts on ACE2 sequence alone and not as forcefully explain why this is unlikely to be a sufficient proxy (can leave this description until results or discussion), leaving the punchiness of the conclusion that it is indeed not that powerful as a less obvious outcome than is currently presented after the Introduction.

Agreed, we have removed the text about the suitability of ACE2 from the introduction, since this is already covered more extensively in the Discussion section.

2. Toward the question of the sufficiency of ACE2 binding data alone for predicting susceptibility: when evaluating the hierarchy of evidence used in the final collated dataset (where animal-based studies trump heterologous ACE2 cell culture experiments), were there any cases where an ACE2 receptor is known to be sufficient to enable cell entry but the animal itself is not experimentally susceptible? Highlighting any observations of this seems the most direct evidence to the point that ACE2 sequence alone is insufficient to predict susceptibility.

There were no such cases in our dataset, but the limited number of species with data from multiple data types makes it difficult to draw firm conclusions from this (only 20 species have data from both cell culture-based assays and experimental infections/direct observations). It should also be noted that our data collection strategy was not set up to evaluate this, since we did not continue looking for or recording cell culture data if a natural infection or experimental infection study was found first for a given species.

3. I wonder whether a more pseudo-mechanistic two-step model could be considered or proposed in the Discussion – a first step based on ACE2 sequence (the "first step" necessary for susceptibility), and the second based on broader determinants of susceptibility (which would be best captured by phylogeny given complexity of unknown unknowns). The reason this may be helpful is that ACE2 binding can be "flipped" on or off with individual amino acid mutations and therefore more quickly deviate from phylogenetic trends (e.g. due to virus-host arms races in Rhinolophus bats, PMID 32699095, but see point 6 below). The broader determinants of susceptibility involve many complex components from cell biology to physiology, and so of course can not be tractably ascribed to a single gene sequence, but also may be better captured by a phylogenetic scale anyway. Such a two-tiered model could also better accommodate the multi-tiered data that is collapsed into a single training paradigm in the current study – for example, the simpler heterologous ACE2 entry assays could be incorporated only for the "first step" of model evaluation, while the whole-animal susceptibility data could serve as a target for the second step / integrated two-step model. This model would lose the utility of the phylogenetic model as illustrated in Figure 5, in that ACE2 sequence would need to be known, but could better incorporate multi-modal data to improve predictive accuracy.

We agree that such a multi-stage model would be theoretically preferable, and now mention this in the second-to-last paragraph of the discussion. However, at present data for such an approach are lacking, as one would need to obtain robust sample sizes for each individual data source. At present, this is not possible, which is why we could only produce a suitable sample size by combining all data sources.

4. Many species of bats show dramatic ACE2 polymorphism centered on positions contacted by sarbecoviruses, and this variation is known to influence binding of certain sarbecoviruses (e.g., PMID 32699095). How did you account for ACE2 polymorphism in analyses -- did you just resolve each species to a single ACE2 sequence? And how do you incorporate into the model possibilities that certain ACE2 alleles within a single species are permissive to entry by some sarbecoviruses while others are not?

Following the practice of the previous studies we sought to evaluate, susceptibility was modelled at the host species-level. This was necessary given the limited data available and particularly the low numbers of species for which susceptibility was known below the species level. Similar issues prevent us from accommodating below-species-level variability in ACE2 – most species have just one ACE2 sequence available, and including multiple sequences for only some species (e.g., by random sampling in different training rounds) would bias models towards better-studied species (which tend to be better studied because they are susceptible). We now acknowledge these limitations in the discussion.

We have also clarified our choice of representative ACE2 sequences in the methods section, emphasising that these sequences were from a curated database to maximise data quality.

5. Related to the above point (and a concern that I had about species like R. pearsonii which are seemingly not susceptible to ACE2-utilizing sarbecoviruses, but then I saw Figure 2 – supplement 2): given that the dynamics of long-term host:virus coevolution that exist in Rhinolophus bats are so different from the dynamics of susceptibility that are germane to questions of reverse zoonotic and intermediate/amplifying potential of other species – might it be wise to exclude Rhinolophus species from the analyses as they may require different forms of "signal" for prediction of susceptibility compared to the rest of mammals?

The decision tree-based models we use should in theory be robust to such differences, since different groups of species can follow completely different explanation paths through the decision trees if needed. For example, models based on representations of amino acid positions could use only partially-overlapping or even entirely different sets of positions to predict different groups of species (e.g., rhinolophid bats). In practice however, our results do not support the idea that the model uses different amino acid positions to predict rhinolophid bats. This is illustrated by the new inset added to Figure 3 —figure supplement 1, which shows that feature usage by our model does not seem to fundamentally differ between rhinolophid bats and other species. As an extra check, we also followed the reviewer’s suggestion to test our models after excluding *Rhinolophus* species (Figure 3 —figure supplement 3). We find no discernible difference in performance from our main models. These new results have been added to the second paragraph of the discussion, where they further strengthen our argument that ACE2 sequence variation is not required for prediction of susceptibility (at least at the broad, species-level scale investigated here).

Reviewer #2 (Recommendations for the authors):Pg 2 L11-12 – has anyone looked at the correlation between susceptibility in cell culture vs in vivo (for any virus)? Might be complex as often comparing across tissue types as well as species. Some support from https://doi.org/10.1371/journal.ppat.1004475 but not sure if there are other studies such as this looking directly at susceptibility?

We are not aware of any systematic comparisons, but in general there are many viruses which remain difficult to culture even once a susceptible host is clearly known (e.g., Human Coronavirus HKU1, see https://journals.asm.org/doi/10.1128/JVI.00947-10). In our own data there were two conflicting cases, with one native cell culture assay and one heterologous ACE2-based assay failing to establish infection while experimental infections succeeded (for *Mesocricetus auratus* and *Callithrix jacchus*, respectively). However, as discussed in response to point 2 of reviewer 1, our data collection was not designed to systematically investigate the correspondence between different susceptibility assays.

Pg 3 – L23-30 – re the assumption that infection with SARS-CoV suggests susceptibility also to SARS-CoV-2 and vice versa – this seems reasonable, although may expect some instances of virus by species interactions meaning this is not the case.. Looking at the supp data it looks like in the 6 species tested with both SARS^-1^ and -2 they can be infected by both (ie 6/6) – is that correct? If so maybe alter the text to say this specifically, and offer some more general support for your approach with (eg their own recent paper in PNAS or https://onlinelibrary.wiley.com/doi/10.1111/tbed.14361)

The correct number is 5/5 species with data for both SARS-CoV and SARS-CoV-2; the 6th referred to infection by both SARS-CoV-2 and a different sarbecovirus. However, only two of these records include experimental infections, while all five species have reports of natural infection with both viruses. As a result, we do not know the true denominator (i.e., how many species have been tested for susceptibility to both viruses), and therefore prefer the more conservative wording used in the manuscript:

“… among the 10 included host species with records of natural SARS-CoV-2 infection, many were also reported to be susceptible to other sarbecoviruses (N=6), primarily SARS-CoV (N=5, supplementary file 1).”

Pg 4 L3-10 felt some more detail was needed here on ACE2 data to help the reader follow the approach

We have now clarified in the second results paragraph that our dataset consists primarily of NCBI reference sequences, with a more detailed description added to the methods section.

The authors use the described modelling approach to predict what species are susceptible to sarbecovirus infection, aggregated by taxonomic order (Figure 5, Figure Supplement 2). From the figure it is apparent that the observed data set contains primarily mammalian samples where as the predicted dataset contains a considerably higher proportion of avian samples. Given that the narrative around reverse zoonosis is largely focused on mammals, it would be helpful to have more of a discussion around the role of avian species both in these analyses and in the transmission of sarbecoviruses.

Agreed. We now explain in the penulimate paragraph of the results section that while predictions are made available for all birds and mammals with suitable data, our analyses focus on mammals given the small number of training examples available for birds.

Reviewer #3 (Recommendations for the authors):You write"Second, since no model has been trained or validated using observed data on infection (the outcome of interest)"Then later:"Understanding the value of ACE2-based host range predictions for guiding surveillance therefore requires developing models based on the outcome of interest – susceptibility – and quantifying the accuracy of their predictions."And later again:"We treated these different sources of information hierarchically, considering the best available evidence for compatibility or incompatibility in each host species (natural infection > experimental infection > cell culture > heterologous ACE2 cell culture experiments)."I'd recommend changing "infection" to susceptibility in the first-quoted passage. To me (maybe just me?) "infection" makes me think infection of a live animal, even though I know that cells in culture can be infected too.

Agreed. This has been changed.

We also now mention the issue of new variants potentially having different host ranges raised in the public review as a limitation in the discussion.